# The Robust Self-Assembling Tubular Nanostructures Formed by gp053 from Phage vB_EcoM_FV3

**DOI:** 10.3390/v11010050

**Published:** 2019-01-11

**Authors:** Eugenijus Šimoliūnas, Lidija Truncaitė, Rasa Rutkienė, Simona Povilonienė, Karolis Goda, Algirdas Kaupinis, Mindaugas Valius, Rolandas Meškys

**Affiliations:** 1Department of Molecular Microbiology and Biotechnology, Institute of Biochemistry, Life Sciences Centre, Vilnius University, Saulėtekio av. 7, LT-10257 Vilnius, Lithuania; lidija.truncaite@bchi.vu.lt (L.T.); rasa.rutkiene@bchi.vu.lt (R.R.); simona.poviloniene@bchi.vu.lt (S.P.); rolandas.meskys@bchi.vu.lt (R.M.); 2Sector of Microtechnologies, Institute of Biotechnology, Life Sciences Centre, Vilnius University, Saulėtekio av. 7, LT-10257 Vilnius, Lithuania; karolis.goda@gmail.com; 3Proteomics Centre, Institute of Biochemistry, Life Sciences Centre, Vilnius University, Saulėtekio av. 7, LT-10257 Vilnius, Lithuania; algirdas.kaupinis@gf.vu.lt (A.K.); mindaugas.valius@bchi.vu.lt (M.V.)

**Keywords:** self-assembly, nanotubular structures, tail sheath protein, bacteriophage vB_EcoM_FV3

## Abstract

The recombinant phage tail sheath protein, gp053, from *Escherichia coli* infecting myovirus vB_EcoM_FV3 (FV3) was able to self-assemble into long, ordered and extremely stable tubular structures (polysheaths) in the absence of other viral proteins. TEM observations revealed that those protein nanotubes varied in length (~10–1000 nm). Meanwhile, the width of the polysheaths (~28 nm) corresponded to the width of the contracted tail sheath of phage FV3. The formed protein nanotubes could withstand various extreme treatments including heating up to 100 °C and high concentrations of urea. To determine the shortest variant of gp053 capable of forming protein nanotubes, a set of N- or/and C-truncated as well as poly-His-tagged variants of gp053 were constructed. The TEM analysis of these mutants showed that up to 25 and 100 amino acid residues could be removed from the N and C termini, respectively, without disturbing the process of self-assembly. In addition, two to six copies of the gp053 encoding gene were fused into one open reading frame. All the constructed oligomers of gp053 self-assembled in vitro forming structures of different regularity. By using the modification of cysteines with biotin, the polysheaths were tested for exposed thiol groups. Polysheaths formed by the wild-type gp053 or its mutants possess physicochemical properties, which are very attractive for the construction of self-assembling nanostructures with potential applications in different fields of nanosciences.

## 1. Introduction

Over the past decades, the construction of various nanostructures, based on self-assembling biomolecules, is of special interest for both an understanding of the fundamental point of view as well as due to the enormous potential for application in industry, medicine and many other areas [1,2,3]. Self-assembling structures open up exciting opportunities for the development of various tools, including biosensors, energy storage devices, drug delivery systems and nanobiopolymeric scaffolds [4,5,6,7]. Therefore, it is not surprising that viruses, including bacteriophages, have been used for the preparation of nanoscaled materials [8,9,10,11,12,13,14,15].

Filamentous bacteriophage M13 and its relatives are the phages of choice, most often used for the construction of genetically engineered viruses, which have been adapted to build phage-based nanosensors, liquid crystals and films [16,17], micro- and nanofibers [14,18], as well as having use in tissue regeneration and to build other functional materials [19,20,21,22,23], or to fabricate nanorings [24]. The construction of self-assembling structures, made from the genetically engineered recombinant proteins of tailed bacteriophages, including podoviruses P22, phi29 [25,26] and myovirus T4 [4,27,28], have also been reported.

During the morphogenesis process of the tailed bacteriophages, the assembly pathways of the structural components follow a strict order and consist of many steps where the subsequent attachment of proteins is controlled by different mechanisms [29,30,31]. Thus, tail sheath proteins alongside tail tube proteins constitute a major part of the contractile tail of myoviruses [31,32]. The polymerisation of the tail sheath proteins, which assemble around the tail tube, starts at the tail tip complex (the baseplate) and is propagated to the end of the tube. The sheath subunits arrange into a stack of hexameric rings, rotated relative to each other, thus creating a 6-start helix [33,34]. During the process of virion assembly inside the cell, the length of the tail sheath is determined by the length of the tail tape measure protein, which is used as a scaffold for the polymerisation of both tail tube and tail sheath proteins. When the sheath reaches the length of the tube, the tail terminator protein binds to the tail tube terminator protein and the last row of the tail sheath subunits to complete the tail, which then becomes available for attachment to the head [31,33].

It has been demonstrated that, in the absence of the baseplate or the tail tube, the recombinant tail sheath proteins of bacteriophages self-assemble both in vivo and in vitro into tubular structures of variable lengths called polysheaths which have the same helical parameters as the contracted tail sheath [35,36,37,38]. The polysheath is a stable structure, which withstands treatment with various chemical and physical factors [39,40,41]. A number of deletion mutants of tail sheath proteins have also been constructed and analysed [29,38,42,43].

In order to expand the current knowledge on biotechnologically attractive self-assembling structural proteins from phages, a tail sheath protein (gp053) of *E. coli* infecting myovirus vB_EcoM_FV3 (FV3) [44] was studied in detail. We showed that the recombinant gp053 self-assembled into long ordered and very stable polysheaths. In addition, the gp053 mutants harbouring the deletions at the N and/or C terminus, with or without the His-tag, as well as the genetically fused homooligomers of gp053 were constructed and their tendency to form nanostructures was analysed.

## 2. Materials and Methods

### 2.1. Cloning Procedures

The PCR fragment of gene *053* (GeneID: 14011712) from phage FV3 was obtained by amplification of the phage DNA using the primers presented in Appendix A. The purified PCR products were cleaved with NdeI and BamHI/XhoI (Thermo Fisher Scientific, Vilnius, Lithuania) and then inserted into the pET-16b or pET-21a (Novagene, Madison, WI, USA) vectors, digested with the appropriate restriction endonucleases. The resulting vectors were amplified in E. coli DH10B (Invitrogen, Dublin, Ireland) cells and verified by DNA sequencing. These vectors were used for the production of a recombinant full-length gp053 and the truncated gp053 mutants with various variations in length and His-tag (Appendix A). The construction scheme of the gp053 oligomers is presented in detail in Appendix A.

### 2.2. Protein Expression

Protein production was carried out in the *E. coli* strain BL21 (DE3) (Novagene, Madison, WI, USA). The 15 ml of cells that were transformed with the appropriate plasmid were grown at 37 °C to an OD_600_ of 0.5, induced with 0.1 mM IPTG and then incubated at 30 °C for 3 hours. The cells were harvested by centrifugation at 4000× *g* for 5 min at 4 °C, resuspended in 1.5 mL TE (20 mM Tris–HCl (pH 7.8)) and 1 mM EDTA, 4 °C) buffer solution and then disrupted by sonication. The cells with expressed His-tagged recombinant gp053 proteins, which were further purified by using a metal-chelating sorbent, were resuspended in 1.5 mL His-Wash (50 mM sodium phosphate buffer (pH 7.7)), 300 mM NaCl, 50 mM imidazole and 0.03% Triton X-100, 4 °C) buffer solution. The cells were sonicated on ice, in 2.0 mL tubes at 30% amplitude for 5 min of total ON time (30 s on/30 s off) by using the Bandelin SonoPuls HD 2070 homogeniser (BANDELIN, Berlin, Germany). The crude extracts were centrifuged at 4 °C for 15 min at 21,000× *g* to remove cell debris. The cell-free extracts and pellets were directly analysed by SDS-PAGE as well as by TEM.

### 2.3. Protein Purification

The polysheaths formed by the recombinant gp053 were precipitated from the supernatant by the addition of ammonium sulphate to a final concentration of 10%. After incubation for 10 min on ice and centrifugation at 9000× *g* for 15 min at 4 °C, the supernatant was removed, and the pellet was suspended in TE buffer (1/10 of the initial volume) and stored at 4 °C. The shorter, less ordered polysheaths were purified by the addition of ammonium sulphate to a final concentration of 15–20% and/or by repeated centrifugation at 9000× *g* for 15 min at 4 °C. Alternatively, the recombinant His-tagged protein purification using the metal-chelating sorbent was performed by using the His-Spin Protein Miniprep kit (Zymo Research, Irvine, CA, USA) according to the manufacturer’s recommendations. The concentration of the recombinant protein was determined by using a method described by Lowry et al. [45].

### 2.4. Limited Proteolysis of Recombinant Proteins with Trypsin

Trypsin (2 mg/mL) was added into the protein solution (~2 mg/mL) to form a final protease:protein ratio of 1:100 (*w*/*w*), and the solution was then incubated at 22 °C. Aliquots (10 μL) were withdrawn from the reaction mixture at different time intervals and mixed with 10 μL of SDS sample buffer (2× concentrated). The reaction was stopped by heating the solution in boiling water for 5 min. The samples were analysed by electrophoresis in SDS-polyacrylamide gel.

### 2.5. In-Gel Protein Digestion for Mass Spectrometry Analysis, Liquid Chromatography and Mass Spectrometry

In-gel trypsin digestion was performed according to the protocol described by Hellman et al. [46]. LC–MS data were collected as described previously [47]. Briefly, the liquid chromatographic analysis was performed in a Waters Acquity ultra performance LC system (Waters Corporation, Wilmslow, UK). The peptide separation was performed on an Acquity UPLC HSS T3 250 mm analytical column. The MS data were acquired using the Synapt G2 mass spectometer and Masslynx 4.1 software (Waters Corporation) in positive ion mode using data independent (DIA) acquisition (MSE).

### 2.6. Labelling of gp053 Polysheaths with Neutravidin-Conjugated Gold Nanoparticles

The labelling of polysheaths with gold nanoparticles was carried out according to the published procedure [48] with some modifications. The suspension of the purified recombinant gp053 was treated with 1 mM of tris(2-carboxyethyl)phosphine (TCEP) (AppliChem, Darmstadt, Germany) to reduce possible disulphide bonds. The reduced protein was dialysed against PBS (pH 7.5) for 1 h at 22 °C. After 1 hour of biotinylation at 22 °C, the mixture was dialysed against PBS (pH 7.5) buffer. The efficiency of biotinylation was evaluated and the biotinylated polysheaths were incubated with neutravidin-conjugated 10 nm gold nanoparticles (Nanopartz, Loveland, CO, USA) as described previously [48]. The mixtures of polysheaths and nanoparticles were incubated for 1 to 5 days at 4 °C. The incubation of non-biotinylated polysheaths with neutravidin-conjugated 10 nm gold nanoparticles was carried out under the same conditions.

### 2.7. Transmission Electron Microscopy (TEM)

The images of gp053 were obtained by transmission electron microscopy of the negatively-stained samples. Approximately 10 µL of the diluted sample solution was directly applied on the carbon-coated nickel grid (Agar Scientific, Essex, UK), and the excess liquid was drained with filter paper before staining with two successive drops of 2% uranyl acetate (pH 4.5). The prepared sample was dried and examined with a Morgagni 268(D) transmission electron microscope (FEI, Hillsboro, OR, USA).

### 2.8. Bioinformatics and Molecular Modeling

The bioinformatics analysis of gp053 was performed using the Fasta-Nucleotide, Fasta-Protein, BLASTP, Transeq [49] and Clustal Omega [50]. The phylogenetic analysis was conducted using MEGA version 5 [51]. The Protein Information Resource (PIR) server was used for calculating the predicted molecular mass of the recombinant protein [52]. The search for the gp053 fold was conducted using the HHpred server [53,54,55]. The I-TASSER server [56,57] was used to model the whole gp053. UCSF Chimera [58,59] was used for the visualisation and analysis of the predicted molecular structure of gp053.

## 3. Results

### 3.1. Bioinformatics Analysis

Based on the results of the bioinformatics analysis, the tail sheath protein encoded by gene *053* (GeneID: 14011712) of the enterobacteria phage FV3 [44] had the closest identity (96–99%) to the tail sheath proteins from the eleven *Escherichia* infecting bacteriophages, which belonged to the genus *V5virus* within the subfamily *Vequintavirinae* (Appendix A).

The HHpred analysis of the amino acid sequence of gp053 revealed that it corresponded to the fold of the two tail sheath proteins of bacteriophages. Hence, the residues 8 to 442 of gp053 were predicted to adopt the fold of the tail sheath protein from *Staphylococcus* phage phi812 (PDB ref 5LI4) with a probability of 97.05 (*E*-value, 0.047), whereas C-terminal fragment of gp053 (residues 258 to 442) was predicted to adopt the fold of the C-terminal fragment (residues 470 to 643) of the tail sheath protein, gp18, of the phage T4 (PDB ref 3J2M) with a probability of 97.2 (*E*-value, 3.4 × 10^−4^). In addition, the HHpred analysis revealed that the predicted fold of gp053 was similar to the contractile phage-like structures from bacteria. The residues 1 to 441 of gp053 were predicted to adopt the fold of the R-type pyocins from *Clostridium difficile* (PDB ref 6GKW) and *Pseudomonas aeruginosa* (PDB ref 3J9Q) with a probability of 99.51 (*E*-value, 1.2 × 10^−14^) and 98.6 (*E*-value, 9.9 × 10^−7^), respectively. The residues 18 to 442 were aligned with the tail sheath protein encoded by gene *lin1278* from the prophage infecting *Listeria innocua* (PDB ref 3LML) with a probability of 98.5 (*E*-value, 1.3 × 10^−5^). Furthermore, the residues 16 to 442 were aligned to the tail sheath protein encoded by gene *dsy3957* from the prophage of *Desulfitobacterium hafniense* (PDB ref 3HXL) with a probability of 98.5 (*E*-value, 6.3 × 10^−6^).

The best 3D model of the full-length gp053 predicted using the I-TASSER server (Figure 1) showed a C-score of –1.77 and a TM-score of 0.50 ± 0.15. The top three proteins from the PDB that had the closest structural similarity to the predicted model were 3LML, 3HXL and 6GKW with a TM-score of 0.801, 0.659 and 0.602, respectively.

### 3.2. Production and Analysis of the Recombinant gp053

Initially, for gene *053* expression in *E. coli* cells, three plasmids encoding wild-type, N- and C-His-tagged gp053 were designed (Appendix A). The soluble recombinant gp053 with a molecular mass of ~50 kDa (predicted mass—50.245 kDa) was produced after induction with IPTG and incubation at 30 °C for three hours. The TEM analysis of the cell-free extracts revealed that the overexpressed gp053 self-assembled into regular tubular structures—polysheaths (Figure 2C). The width of the gp053 polysheaths (27.13 ± 2.69 nm) corresponded to the width of the contracted tail (26.79 ± 1.78 nm) of the phage FV3 [44]. The length of the polysheaths varied from separate rings with an internal hole of 11.51 ± 0.78 nm and a diameter of 27.83 ± 2.68 nm to nanotubes up to 1000 nm long. The attempts to purify long ordered His-tagged gp053 polysheaths effectively by using the metal-chelating sorbent were unsuccessful (Figure 2). The SDS-PAGE analysis revealed that the vast majority of the recombinant proteins did not adsorb onto the sorbent. Moreover, according to the results of the TEM analysis, the affinity-purified recombinant gp053 formed noticeably shorter tubular structures (Figure 2D,E).

Alternatively, we have demonstrated that the purification of the polysheaths could be performed quickly and effectively by using ammonium sulphate precipitation (Figure 2F, Figure 3). The yield of the bacteria-derived and purified structures was 3.75 ± 0.35 mg/g of wet cells. The TEM analysis of the purified protein showed that the morphology of the polysheaths (Figure 3B) corresponded to the morphology of the tubular structures observed in the cell-free extracts.

The purified polysheaths exhibited extraordinary stability. The structures withstood overnight incubation in 8 M urea. Moreover, even boiling them for 30 min or prolonged storage (more than one year) in TE buffer at 4 °C did not affect the morphology of the gp053 polysheaths (Figure 4C,E,F). Also, the polysheaths remained as ordered tubular structures after incubation with trypsin (Figure 4D).

The detailed analysis of the trypsin-treated protein tubes by SDS-PAGE showed that fragments with molecular masses of approximately 43, 39 and 37 kDa appeared in addition to the full-length gp053 (Figure 5). Based on these results, it was concluded that only several trypsin digestion sites out of the 22 existing in the gp053 protein were available on the surface of the polysheaths. To identify those sites, the protein bands were excised from the gel before and after proteolysis by trypsin, purified according to the protocol described by Hellman et al. [46], and liquid chromatography coupled with mass spectrometry (LC–MS/MS) was performed. The protein sequence coverage map was obtained from the ProteinLynx Global Server (PLGS) processed MS^E^ data (three technical replicates) (Appendix A). The results of the qualitative and quantitative MS analysis of the peptides showed that the vast majority of the proteins in the suspension of the purified polysheaths consisted of the potential full-length gp053 (Figure 5B, lane 1, arrow 1), whereas a minor protein band (Figure 5B, lane 1, arrow 2) contained gp053 with a potential 86 N-terminal amino acid truncation. This truncated protein was expected to occur due to the proteolytic activity of some of the proteases from the *E. coli* cells, since the protease inhibitors were not added during the purification of the polysheaths. After treatment with trypsin, the most intensive protein bands were identified as potentially full-length gp053 (Figure 5B, lane 2, arrow 3), gp053 with the C-terminal truncation of 41 amino acids (Figure 5B, lane 2, arrow 4) and gp053 with the C-terminal truncation of 77 amino acids (Figure 5B, lane 2, arrow 5). The minor protein band consisted of the gp053 with potential truncations of 86 N-terminal and 77 C-terminal amino acids (Figure 5B, lane 2, arrow 6).

According to this analysis, trypsin digestion of gp053 proceeded in a specific manner and resulted in the potential cleavage of the peptide bond between Lys381–Asn382 and Arg417–Val418 (Figure 1).

### 3.3. Construction of gp053 Mutants

In order to obtain truncated recombinant proteins still able to form regular, stable tubular structures and to understand the polymerisation properties of gp053, we constructed a set of the gp053 mutants truncated at the N or/and C terminus. Those recombinant proteins corresponded to the full-length gp053 from 98.0% (gp053_NΔ9) to 43.7% (gp053_CΔ200) (Appendix A, Figure 6).

The recombinant proteins were produced in *E. coli* BL21 (DE3) cells and the cell-free extracts were analysed by TEM. Whenever an assembly into the polysheaths was observed, the purification procedures using ammonium sulphate were performed. The TEM analysis revealed that deletions of 9, 20 and 25 amino acids from the N-terminal region did not disturb the protein assembly into polysheaths. Meanwhile, a deletion of 29 amino acids from the N-terminal region abolished the ability of the truncated gp053 to assemble into the typical polysheaths. Therefore, the gp053_NΔ29 was found in a soluble fraction, where it was folded into the unordered protein ribbons (Appendix A, Figure 6). On the other hand, the deletions of 11, 31, 51, 76 and even 100 amino acids from the C terminus did not abolish the formation of the ordered tubular polysheaths. However, the gp053_CΔ150 mutant lacking 152 amino acids at the C terminus formed insoluble aggregates in *E. coli* cells (Appendix A, Figure 6).

The TEM analysis of the polysheaths formed by the majority of the mutants, except for gp053_N-his and gp053_CΔ100, revealed that the structures showed no significant differences compared to the polysheaths formed by the wild-type recombinant gp053 (Appendix A, Figure 7).

On the contrary, the polysheaths formed by the gp053_N-his mutant were less ordered and approximately 2–5-fold shorter in length (~100–300 nm) (Figure 7A). In the case of the gp053_CΔ100, the tubular structures were longer (up to ~3000 nm) and slightly larger in diameter (~32.47 nm) (Figure 7H,I) compared to the wild-type ones. In addition, it was observed that some of the nanotubes, for example, formed by gp053_NΔ20, gp053_CΔ76 and gp053_CΔ100 (Figure 7F–H, respectively), contained more clearly defined internal channels. A similar morphology was also detected in the case of the full-length gp053 after prolonged storage in TE buffer or treatment with trypsin (Figure 4C,D, respectively).

### 3.4. Investigation of the Oligomeric Constructs of gp053

According to the 6-start helix symmetry of the polysheath, the gp053 oligomers were genetically engineered (Appendix A, Figure 8A) and their ability to form the ordered structures was analysed. Initially, it was found that all the recombinant proteins (gp053_N_C_mon, gp053_N_C_dim, gp053_N_C_trim, gp053_N_C_tet, gp053_N_C_pent and gp053_N_C_hex) were soluble under the investigated conditions (Figure 8C).

When the samples were analysed by TEM immediately after the disruption of cells, the ordered tubular structures up to ~400 nm in length were visible only in the case of the monomeric gp053. Some tubular structures up to ~200 nm in length were also visible in the case of the dimeric gp053. The other oligomers of gp053 formed indefinite structures (Figure 8D). However, the tubular structures, although of a less regular structure, were observed in the case of all oligomers after an incubation of the samples in TE buffer with periodic shaking at 22 °C for 48 h (Figure 8D).

### 3.5. Labelling of gp053 Polysheaths with Neutravidin-Conjugated Gold Nanoparticles

Based on the model of the structure of gp053, one of the four cysteines, Cys160, was located on putative domain 2 (Figure 1), which was found on the outer surface of the polysheaths formed by LIN1278, DSY3957 and gp18 [38]. In order to determine whether this amino acid was on the surface of the nanotubes formed by recombinant gp053, the purified polysheaths were modified with biotin and incubated with neutravidin-conjugated gold nanoparticles. The TEM analysis revealed that the gold nanoparticles in most cases were visible on the ends or breakpoints of the polysheaths, single rings or short gp053 aggregates, and the rest of the outer surface of the protein nanotubes was not decorated with gold nanoparticles (Figure 9B).

Thus, we can conclude that the cysteine residues of gp053 were buried inside of the protein nanotubes, and only could be accessible for labelling with neutravidin-conjugated gold nanoparticles at the termini of the polysheaths.

## 4. Discussion

The construction of various nanostructures based on self-assembling biomolecules is currently of special interest due to the potential for diverse application in different fields of nanobiosciences. However, only a limited number of self-assembling proteins from bacteriophages have been studied in detail to date. Therefore, in this study, we aimed to shed more light on the diversity of self-assembling tubular nanostructures and chose to investigate the tail sheath protein (gp053) from enterobacteria phage vB_EcoM_FV3.

Despite 98–99% sequence identity of the tail sheath proteins from phages of the genus *V5virus*, the molecular architecture of FV3 gp053 is largely undetermined. To our knowledge, none of the crystal structures of the tail sheath proteins from the genus *V5virus* or their close relatives have been resolved to date. On the other hand, it has been reported that the structure of the domains constituting the tail sheath proteins from phages T4, phiKZ, and phi812 are similar despite lower than 15% sequence identity between the aforementioned proteins [29,60,61]. It has also been shown that contractile structures from phages have very similar architecture to contractile molecular machines found in many prokaryotes: R-type pyocins, the Type VI secretion system (T6SS) and phage tail-like protein translocation structures (PLTS) [32,62,63,64,65]. Moreover, phylogenetic analysis of the tail sheath proteins DSY3957 (PDB ref: 3HXL) and LIN1728 (PDB ref: 3LML) from the prophages revealed that, phylogenetically, they are more closely related to the R-type pyocin from *Clostridium difficile,* than to the tail sheath proteins from other phages [63]. Therefore, it is unsurprising that the HHpred analysis showed the predicted fold of gp053 from phage FV3 to be more similar to the R-type pyocins from *Clostridium difficile* (PDB ref 6GKW) and *Pseudomonas aeruginosa* (PDB ref 3J9Q) than to the fold of the tail sheath proteins from prophages (PDB ref 3LML, PDB ref 3HXL) or phages (PDB ref 5LI4, PDB ref 3J2M). On the other hand, bioinformatics analysis revealed that the gp053 identity compared to the previously mentioned proteins at the amino acid level ranged only from 14.6% (3HXL) to 18.8% (6GKW). Therefore, it has been demonstrated that despite the fact that the tail sheath proteins of different *Myoviridae* phages appeared to have similar helical parameters and functioned in a similar manner [29,31,32,38,66,67,68,69,70], completely different sequences of amino acids forming the tail sheath proteins could exist.

Here, we demonstrated that it was possible to obtain comparatively high amounts of recombinant gp053 protein quickly and efficiently through the use of *E. coli* expression systems. It has been shown that recombinant gp053 formed soluble high molecular weight polysheaths during production in the *E. coli* cells, and the purification of these nanostructures could be performed very quickly, cheaply and efficiently using an ammonium sulphate precipitation from the cell-free extracts only (Figure 2). In contrast, the precipitation of the polysheaths of phage PaBG by using ammonium sulphate was not as effective. Therefore, additional fractionation by ultracentrifugation was needed [71]. Additional purification procedures, including ultracentrifugation, hydroxyapatite and anion-exchange chromatography, have been used to purify nanostructures formed by the recombinant tail sheath proteins of phages T4 and phiKZ [38,42].

Another important characteristic of the polysheaths formed by gp053 is their extreme stability towards various physical and chemical treatments. Similarly, the polysheaths consisting of the tail sheath proteins of other phages have also been shown to be very stable structures. It has been demonstrated that the polysheaths of phage T4 retained their structure after incubation in 8 M urea (pH 7.6–12.2), 6 M guanidine hydrochloride (pH 5.6) and 7.8 M acetic acid (pH 1.8–6.1) [39]. Moreover, the polysheaths of phages T4 and phiKZ were also found to be extremely resistant to proteolysis by trypsin [38,41]. However, although the polysheaths of phage phiKZ remained uncrumbled after incubation with trypsin, the protein was completely cleaved into two fragments, as indicated by the disappearance of the full-length protein band and the appearance of two smaller fragments with molecular masses of about 60 and 15 kDa [38]. In contrast, a significant amount of the recombinant gp053 from phage FV3 remained undigested even after 24 hours of incubation with trypsin (Figure 5). It is likely that different trypsinisation profiles can be concerned with the morphological differences of polysheaths of the previously mentioned phages. In the case of the recombinant tail sheath protein of phage phiKZ, polysheaths with more visible internal channels have been observed. It has been suggested that these structures were caused by less compact packing of protein subunits that facilitates stain penetration into the internal channel of the polysheaths [38]. Meanwhile, the polysheaths composed of recombinant gp053 were different. It is possible that these structures are formed of highly compact protein subunits which are difficult for trypsin to reach. On the other hand, polysheaths with a better defined internal channel were observed in the cases of gp053 mutants such as gp053_NΔ20, gp053_CΔ76 and gp053_CΔ100 as well as a full-length gp053 after prolonged storage in TE buffer or after treatment with trypsin (Figure 4 and Figure 7). Therefore, it is likely that less compact structures were formed in these cases.

LC–MS^E^ analysis of the trypsinisation products of the recombinant wild-type gp053 of FV3 revealed the potential trypsin-accessible cleavage sites between Lys381–Asn382 and Arg417–Val418 (Appendix A). Based on the I-TASSER generated model of gp053, Lys381–Asn382 and Arg417–Val418, together with its N and C termini, are found in domain IV (Figure 1), which is structurally similar to the domains of the prophage tail sheath proteins DSY3957 and LIN1278, which have the same fold as the corresponding region (domain IV) of bacteriophage T4 [38]. It was demonstrated that the N and C termini of the tail sheath proteins are found close to each other and form a domain, which is located in the inner part of the contracted sheath [29,30,31,32,38]. Thus, with reference to the conserved structural model of the tail sheath proteins of bacteriophages and contractile molecular machines found in prokaryotes [65], it is the most likely that the N and C termini, as well as Lys381–Asn382 and Arg417–Val418, are located in the inner part of the polysheaths formed by gp053 of FV3. The results of the purification procedures of recombinant gp053 maintained these speculations—attempts to purify long tubular His-tagged gp053 polysheaths effectively by using the metal-chelating sorbent were unsuccessful (Figure 2).

The results of the experiments with FV3 gp053 deletion mutants were in accordance with previous observations that the elimination of the C-terminal residues of tail sheath proteins (gp18) from phage T4 had a less negative effect on the polymerisation properties, compared to deletions of the N terminus [42,43,72]. On the other hand, it has been demonstrated that some T4 gp18 mutants assembled into thinner filaments called ‘‘noncontracted polysheaths’’ (NCPs) [42], which might correspond to the transitional helices described previously [73]. However, NCPs in the case of gp053 mutants were not observed. In contrast, it has been demonstrated that the tubular structures, formed by gp053_CΔ100, were even longer and of a larger diameter than the polysheaths formed by wild-type recombinant gp053 (Figure 7).

In this study, for the first time, to our knowledge, the genetically engineered homooligomers of the tail sheath proteins from bacteriophages have been constructed. It was demonstrated that all of the recombinant gp053 duplicates were soluble under the investigated conditions. Moreover, a self-assembling process of a number of these proteins *in vitro*, in the absence of other viral proteins, was observed (Figure 8). Hence, the tubular, although irregular structures, were visible not only in the case of recombinant gp053 monomer (gp053_N_C_mon) but also and in the cases of other gp053 oligomers including gp053 pentamer (gp053_N_C_pent). These results are not only promising for the construction of self-assembling nanostructures with various materials exhibited in specific locations on their surfaces but they also raise a number of issues about the polymerisation processes of the tail sheath proteins of bacteriophages.

The I-TASSER-generated model of gp053 (Figure 1) showed that one out of four cysteine residues found in gp053, Cys160, was located in the predicted domain 2, which was structurally similar to the protruding domains forming the outer surface of contractile tail machines [29,30,31,32,38]. On the other hand, the results of the labelling of gp053 polysheaths with neutravidin-conjugated gold nanoparticles revealed that maleimide-based biotinylation occurred only at the termini of the polysheaths and none of the four cysteine residues (including Cys160) were located on the outer surface of the polysheaths freely available for biotinylation (Figure 9). Despite the fact that the specific cysteine residue (or several of them) that was biotinylated within gp053 is still unknown, a simple method to adhere gold nanoparticles to the termini of the polysheaths was shown. The attachment of gold nanoparticles could be very beneficial for the immobilisation of uniformly oriented nanostructures [74]. Thus, the results of this study suggest novel ways to construct hybrid self-assembling nanostructures, particularly ones appropriately prearranged on surfaces.

It has been demonstrated that self-assembling tubular nanostructures made of peptides, proteins or filamentous bacteriophages have been used for both fundamental studies and practical applications, including in the construction of biosensors, energy storage devices, drug delivery systems or tissue engineering [12,13,14,75,76]. Given the fact that the polysheaths, formed of the wild-type gp053 or its mutants, possess physicochemical properties which are very similar to the properties of self-assembling nanostructures formed of biomolecules mentioned above (well-defined shape and dimensions in the nanoscale, robust, ordered and intrinsically monodisperse structures), it is likely that gp053 polysheaths can be used for the same practical applications as tubular nanostructures made of peptides, proteins or filamentous bacteriophages. For example, the gp053 polysheaths, which are up to 1000 nm in length, ~28 nm in diameter and with an internal hole of ~12 nm, are very attractive tools for the synthesis of inorganic matter to produce nanowires with properties of interest to energy sciences and the electronic industry. Polysheaths can potentially be used for mineralisation, both in their interior channel and around their exterior surface as has been demonstrated in the case of the tobacco mosaic virus [77,78]. On the other hand, the polymerisation of recombinant tail sheath proteins is not a precisely controlled process and this still remains the main drawback associated with these nanostructures, hence additional studies are needed to tackle this issue.

Nevertheless, the production and purification of polysheaths, formed from recombinant gp053, is a relatively short, simple and cheap process: only after three hours following induction and after a few hours of purification procedures, it is possible to produce a high yield of purified nanostructures. In contrast, the in vivo or in vivo/in vitro propagation of phage M13 and plant viruses takes several days or several weeks, respectively [79,80,81]. In addition, the polysheaths are released from bacteria by cell lysis, thus, these structures are not strictly limited in either the size of the displayed (poly)peptides or in terms of the hydrophobicity of the polypeptide chain, which was demonstrated as a serious limitation of nascent rod-shaped virions to passage through the host’s exit pore [81].

To summarise, gp053 from phage FV3 self-assembles into ordered tubular structures—polysheaths. The assembly proceeds in vivo and in the absence of other viral proteins. Polysheaths formed of the wild-type gp053 or its mutants possess physicochemical properties which are very attractive for the construction of self-assembling nanostructures with potential applications in different fields of nanosciences.

## Figures and Tables

**Figure 1 viruses-11-00050-f001:**
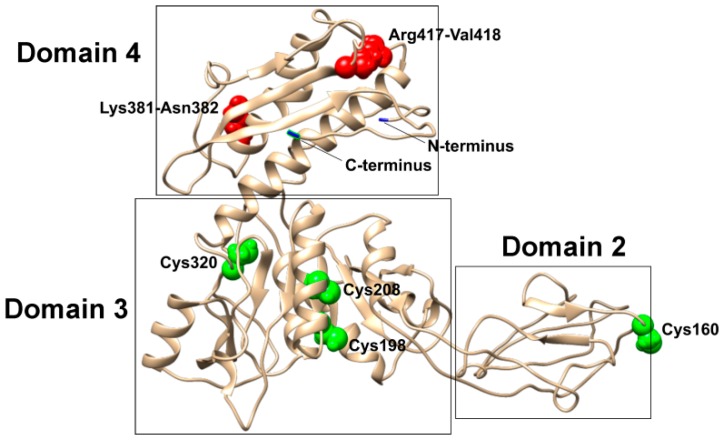
A model of gp053 generated by I-TASSER. The locations of the N and C termini are shown in blue; Cys160, Cys196, Cys208 and Cys320 are shown in green; Lys381-Asn382 and Arg417-Val418 are shown in red. The domain numbering was done according to Kurochkina et al. [38]. The model was visualised using UCSF Chimera.

**Figure 2 viruses-11-00050-f002:**
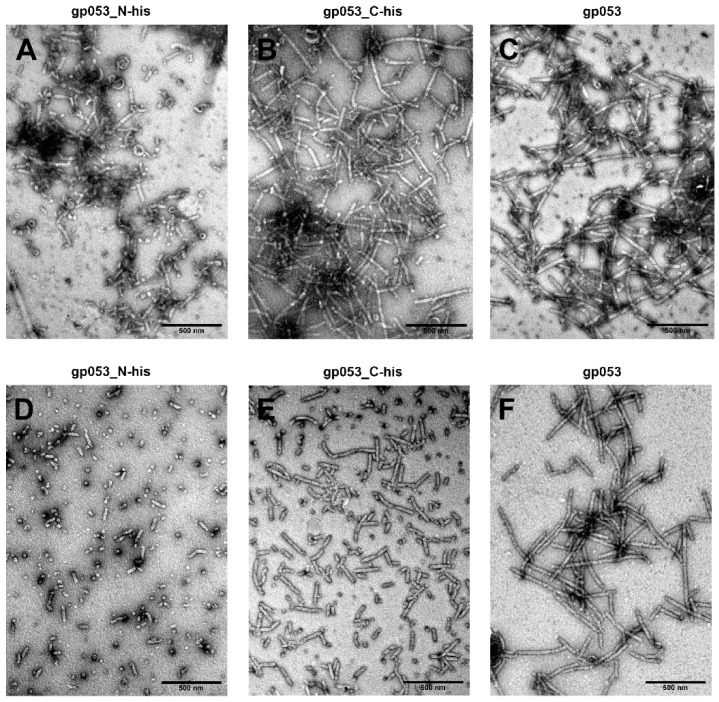
TEM analysis of polysheaths formed by recombinant gp053. The electron micrographs represent the structures before (above) and after (below) purification from the crude cell lysates. The polysheaths formed by recombinant gp053_N-his (**A**,**D**) and gp053_C-his (**B**,**E**) were purified using the His-Spin Protein Miniprep kit (Zymo Research) according to the manufacturer’s recommendations. The polysheaths, formed by recombinant gp053 (**C**,**F**) were purified by precipitation using ammonium sulphate (10% final concentration).

**Figure 3 viruses-11-00050-f003:**
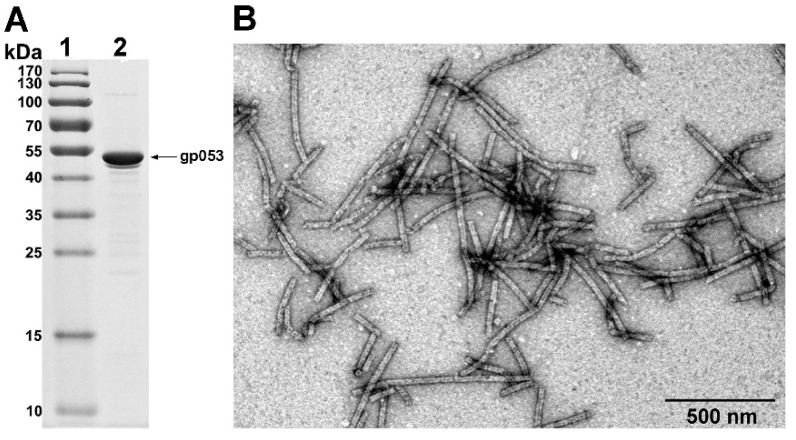
SDS-PAGE (**A**) and TEM (**B**) analysis of the purified wild-type recombinant gp053 produced in *E. coli* BL21 (DE3) cells. Protein synthesis was induced by 0.1 mM IPTG and the cells were incubated at 30 °C for 3 h. The recombinant gp053 was precipitated from the supernatant by the addition of ammonium sulphate to a final concentration of 10%. Lanes: 1—the molecular mass marker, Page Ruler^TM^ prestained Protein Ladder Plus (Thermo Fisher Scientific, Vilnius, Lithuania); 2—the purified wild-type recombinant gp053.

**Figure 4 viruses-11-00050-f004:**
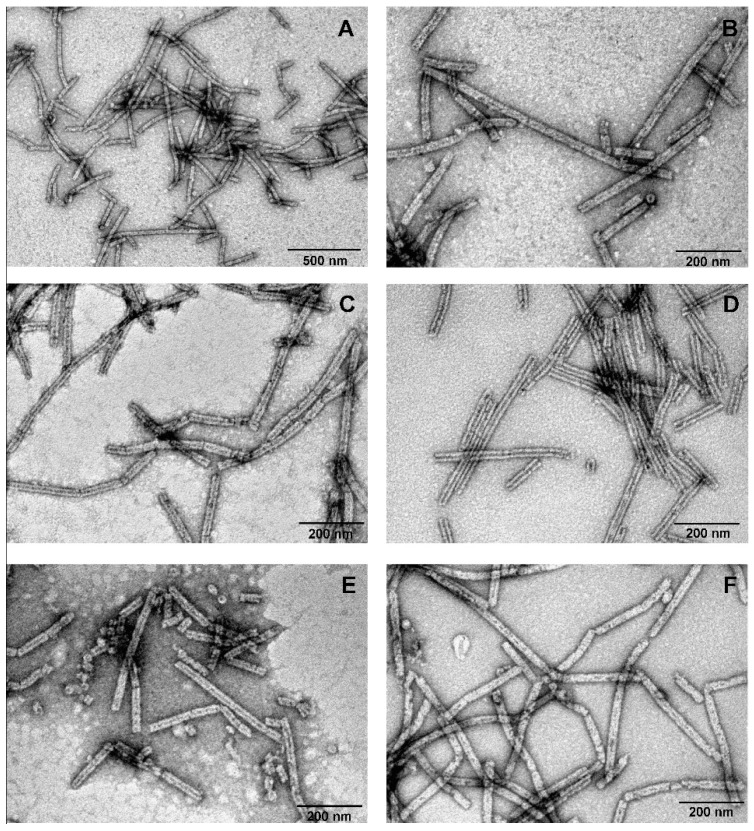
Electron micrographs of the polysheaths formed by wild-type recombinant gp053 from phage FV3. The samples of the purified polysheaths were analysed immediately after purification (**A**,**B**), following storage in TE buffer at 4 °C for 12 months (**C**), after treatment with trypsin (0.02 mg/mL) at 22 °C for 16 h (**D**), after incubation in the presence of 8 M urea at 22 °C for 16 h (**E**), and after incubation in boiling water for 30 min (**F**).

**Figure 5 viruses-11-00050-f005:**
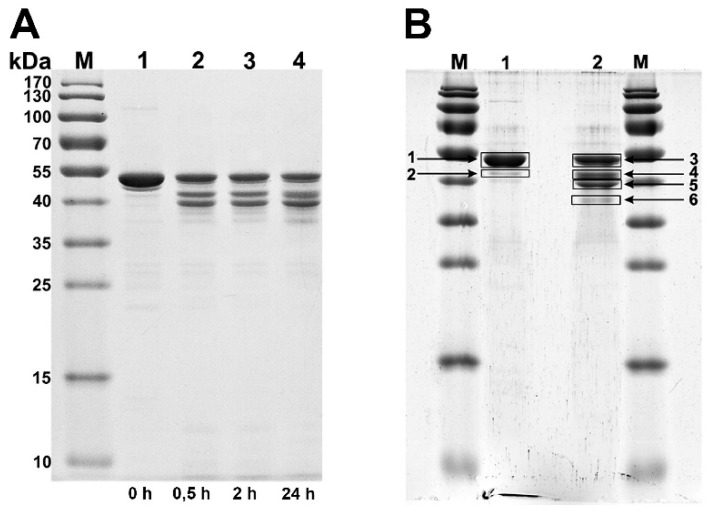
SDS-PAGE analysis of the trypsinisation products of the purified gp053 polysheaths. (**A**) Lanes: M—molecular mass marker, Page Ruler^TM^ Prestained Protein Ladder (Thermofisher), 1—recombinant gp053, 2–4 recombinant gp053 incubated with trypsin (the digestion time is indicated under the lanes). (**B**) Lanes: M—molecular mass marker, Page Ruler^TM^ Prestained Protein Ladder (Thermofisher); 1—recombinant gp053 incubated for 24 hours in TE buffer; 2—recombinant gp053 incubated for 24 hours in TE buffer with trypsin. The arrows indicate the full-length gp053 (1,3) and its fragments (2,4–6), excised from the gel for LC–MS/MS analysis.

**Figure 6 viruses-11-00050-f006:**
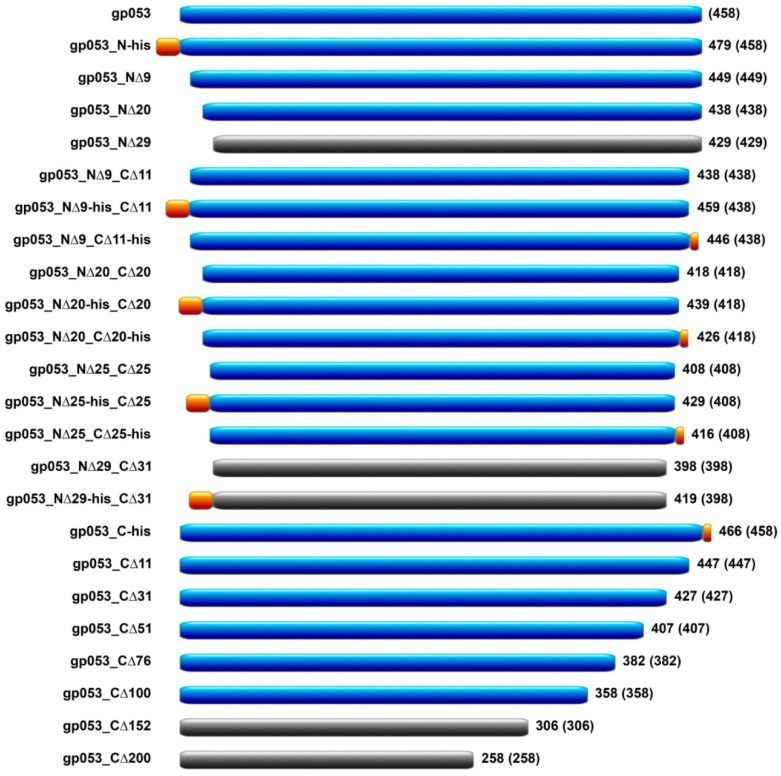
A schematic picture of the truncated mutants of the gp053 protein. The mutants are named (shown on the left) according to N-terminal or C-terminal mutations and the presence of His-tag. The proteins, able to polymerise into nanotubes, are shown in blue. The mutants, incapable of polymerisation, are shown in black. The His-tags are shown in orange. The size of a protein in amino acids is shown on the right, and the portion of wild-type gp053 within a construct is depicted in parenthesis.

**Figure 7 viruses-11-00050-f007:**
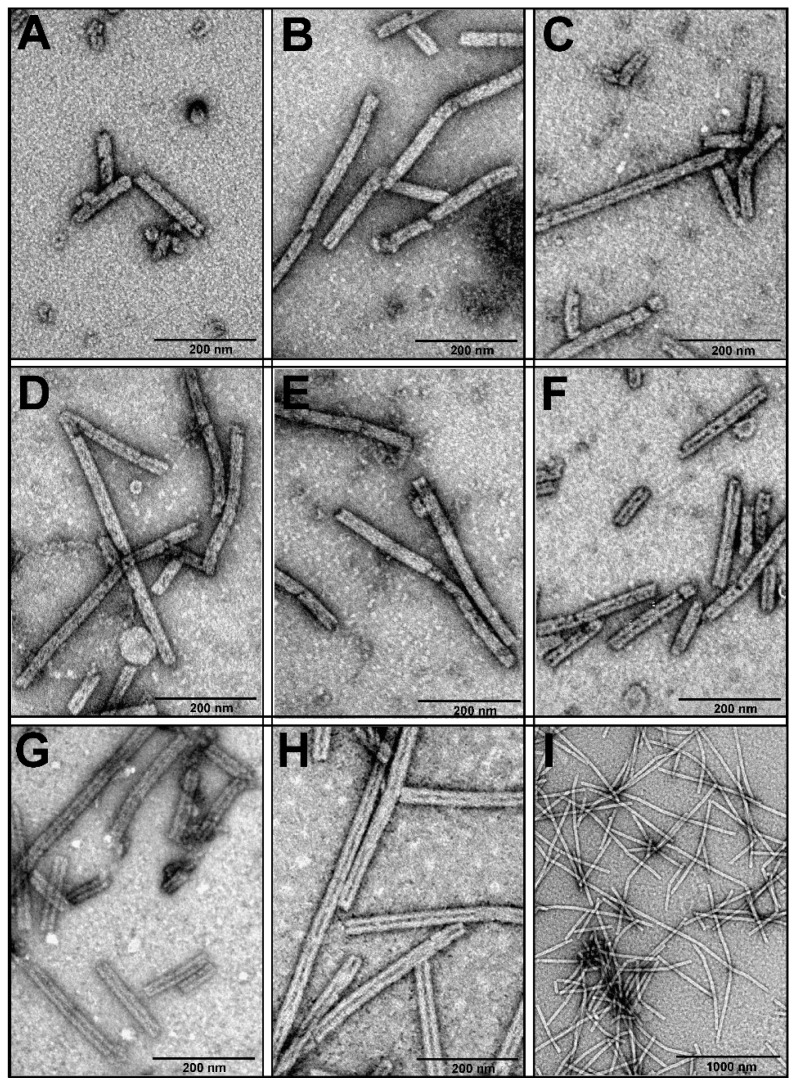
Electron micrographs of the polysheaths formed by the gp053 mutants. (**A**) gp053_N-his, (**B**) gp053_C-his, (**C**) gp053_NΔ9_CΔ11, (**D**) gp053_NΔ9-his_CΔ11, (**E**) gp053_NΔ25-his_CΔ25, (**F**) gp053_NΔ20, (**G**) gp053_CΔ76, and (**H**,**I**) gp053_CΔ100.

**Figure 8 viruses-11-00050-f008:**
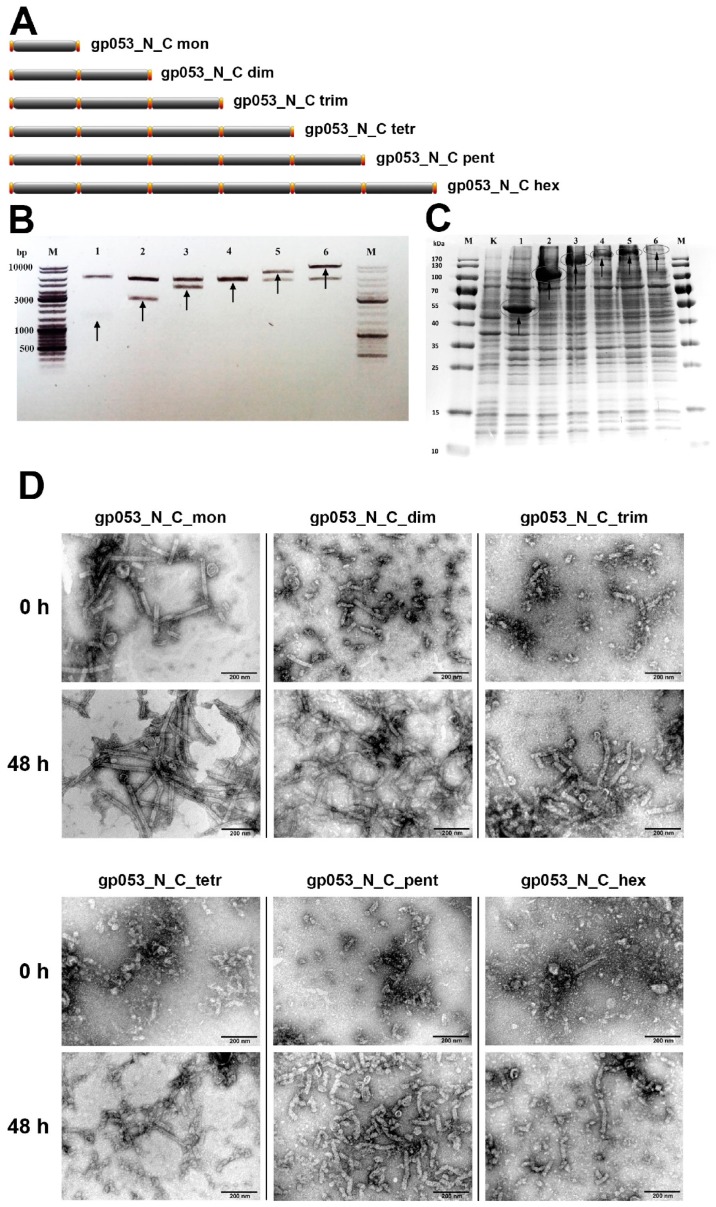
Assembling of the gp053 oligomers. (**A**) A schematic representation of the gp053 oligomeric constructs. The amino acids of the wild-type gp053 within the construct are shown in black, and the insertions with cloning sites are shown in orange. The constructs are named according to the number of copies of the gene *053*. (**B**) The agarose gel electrophoresis analysis of plasmids harbouring the fused *053* genes. The arrows indicate DNA fragments encoding gp053 after hydrolysis with NdeI and XhoI. Lanes: M—molecular mass marker, GeneRuler™ DNA Ladder Mix (Thermo Fisher Scientific, Vilnius, Lithuania); 1—gp053_N_C_mon; 2—gp053_N_C_dim; 3—gp053_N_C_trim; 4—gp053_N_C_tetr; 5—gp053_N_C_pent; and 6—gp053_N_C_hex. (**C**) The SDS-PAGE analysis of the cell-free extracts of the *E. coli* BL21 (DE3) cells producing the recombinant gp053 oligomers. Lanes: M—molecular mass marker, Page Ruler^TM^ prestained Protein Ladder Plus (Thermo Fisher Scientific, Vilnius, Lithuania); K—pET21a (plasmid vector, control); 1—gp053_N_C_mon; 2—gp053_N_C_dim; 3—gp053_N_C_trim; 4—gp053_N_C_tetr; 5—gp053_N_C_pent; and 6—gp053_N_C_hex. (**D**) The TEM analysis of the structures formed by the recombinant gp053 oligomers. The cell-free extracts were analysed immediately after cell disruption and sample preparation or after incubation with periodical shaking at 22 °C. The incubation time is indicated on the left.

**Figure 9 viruses-11-00050-f009:**
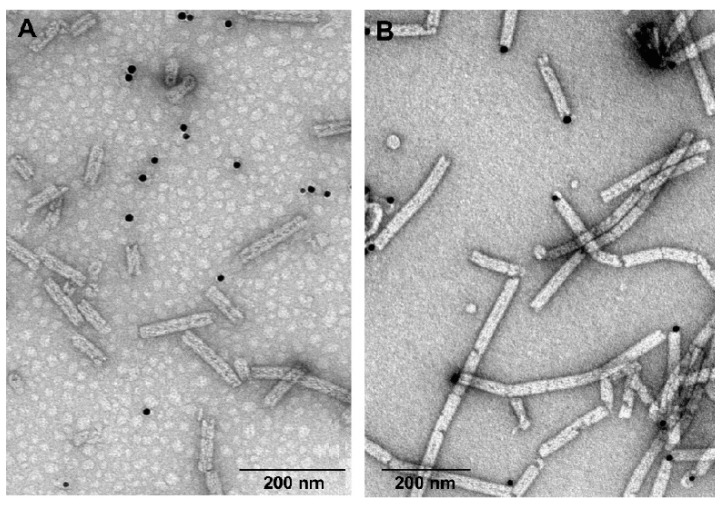
Electron micrographs of the polysheaths formed by the purified recombinant gp053 incubated with the neutravidin-conjugated gold nanoparticles. (**A**) Non-modified gp053, and (**B**) gp053-harbouring cysteines modified with biotin.

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
