# Peer review of "The Robust Self-Assembling Tubular Nanostructures Formed by gp053 from Phage vB_EcoM_FV3"

_viruses, 2019, doi:10.3390/v11010050_

Reviewer 1 Report

Šimoliūnas et al describe production of polysheaths from tail sheath protein gp053 of phage FV3. They used various methods, mainly negative stain electron microscopy, to characterize the polysheaths. In addition they employed mutagenesis to determine core of the protein that is capable of forming the polysheaths and demonstrated that polyproteins containing tandem repeats of gp053 can also assemble. Overall the conclusions of the manuscript are justified by the presented results. However, descriptions in the Materials and methods section are too short and it may be difficult to reproduce the procedures. My major concern is the limited novelty of results presented in the manuscript.

Specific comments:

Line 52, typo “ofa”

Lines 51-53: “During the process of virion assembly inside the cell, the length ofa tail sheath is determined by the length of a tail tube, which is used as a scaffold for the polymerization of the tail sheath protein.”

-   this is misleading, the lengths of both tail tube and tail-sheath are determined by the length of tail tape measure protein.

The section of Materials and methods should include more detail. In the current form the described experiments could not be reproduced.

Materials and methods, section “2.2. Protein expression”

-   the procedure for protein expression is not described in adequate detail. For example, volume of the culture used for induction is not provided, volumes of buffers are not provided, temperatures of buffers, temperatures of centrifugation conditions. Sonication conditions?

-   The use of alternative buffers is not adequate: “Cells were harvested by centrifugation at 4,000 g for 5 minutes, resuspended in TE (20 mM Tris–HCl (pH 7.8)), 1 mM EDTA) or His-Wash (50 mM sodium phosphate buffer (pH 7.7)), 300 mM NaCl, 50 mM imidazole, 0.03% Triton X-100) buffers and disrupted by sonication.”

Lines 100-101: “Trypsin (2 mg/ml) was added into the protein solution in a ratio of 1:100, and incubated at 22°C.” – What was the concentration of substrate?

Results:

Lines 143-148. Tail sheath proteins of phi812 and T4 are related to each other, therefore it seems redundant to state that: "gp053 … corresponded to the fold of two tail sheath proteins of bacteriophages”.

I think the whole section 3.1 is unnecessary and may be removed. The presented analyses take only a few minutes to calculate and lead to no new conclusions.

Lines 174-178: “The attempts to purify His-tagged variants of gp053 by using metal-chelating sorbent were unsuccessful (Supplementary Figure S2). The SDS-PAGE analysis revealed that the vast majority of recombinant protein did not adsorb onto the sorbent. Moreover, according to the results of TEM analysis, the affinity purified recombinant gp053 formed noticeably shorter tubular structures (Supplementary Figure S2D,E).”

-   These two sentences are mutually exclusive. First it is stated that attempts to purify Hos-tagged variants gp053 were unsuccessful. In second sentence it is described that affinity purified gp053 formed shorter tubules.

Author Response

Comments/Authors' response:

Line 52, typo “ofa”

Authors' response

The appropriate changes have been made.

Lines 51-53: “During the process of virion assembly inside the cell, the length ofa tail sheath is determined by the length of a tail tube, which is used as a scaffold for the polymerization of the tail sheath protein.”

-   this is misleading, the lengths of both tail tube and tail-sheath are determined by the length of tail tape measure protein.

 Authors' response

We agree with reviewer's comment. The appropriate changes have been made.

The section of Materials and methods should include more detail. In the current form the described experiments could not be reproduced.

 Materials and methods, section “2.2. Protein expression”

-   the procedure for protein expression is not described in adequate detail. For example, volume of the culture used for induction is not provided, volumes of buffers are not provided, temperatures of buffers, temperatures of centrifugation conditions. Sonication conditions?

Authors' response

We agree with reviewer's comment. The section “2.2. Protein expression” has been amended accordingly.

-   The use of alternative buffers is not adequate: “Cells were harvested by centrifugation at 4,000 g for 5 minutes, resuspended in TE (20 mM Tris–HCl (pH 7.8)), 1 mM EDTA) orHis-Wash (50 mM sodium phosphate buffer (pH 7.7)), 300 mM NaCl, 50 mM imidazole, 0.03% Triton X-100) buffers and disrupted by sonication.”

Authors' response

We agree with reviewer's comment. In Section “2.2. Protein expression” we have made corrections and explained that His-Wash buffer was used only in the case of the cells with expressed His-tagged recombinant gp053 proteins, which were further purified by using metal-chelating sorbent.

Lines 100-101: “Trypsin (2 mg/ml) was added into the protein solution in a ratio of 1:100, and incubated at 22°C.” – What was the concentration of substrate?

Authors' response

The concentration of the substrate was ~2 mg/ml. The appropriate changes have been made in the revised manuscript.

Results:

Lines 143-148. Tail sheath proteins of phi812 and T4 are related to each other, therefore it seems redundant to state that: "gp053 … corresponded to the fold of two tail sheath proteins of bacteriophages”.

Authors' response

It is unclear what did reviewer have in mind by saying “Tail sheath proteins of phi812 and T4 are related to each other”. Identity of these proteins in amino acid level is only 16%. Thus, we do not think that it seems redundant to state that: "gp053 … corresponded to the fold of two tail sheath proteins of bacteriophages”.

I think the whole section 3.1 is unnecessary and may be removed. The presented analyses take only a few minutes to calculate and lead to no new conclusions.

Authors' response

We do not agree with reviewer's comment that section 3.1 is unnecessary one. The results of bioinformatics analysis, presented in this section, provide the reader with additional information about the subject of this study and are used as a background.  In addition, although a model of gp053, generated by I-TASSER and presented in Figure 1, is not completely accurate, it provides a visual information that makes it easier for the reader to understand what is discussed in the following sections. Based on this, the section 3.1 have not been removed from the revised manuscript.

Lines 174-178: “The attempts to purify His-tagged variants of gp053 by using metal-chelating sorbent were unsuccessful (Supplementary Figure S2). The SDS-PAGE analysis revealed that the vast majority of recombinant protein did not adsorb onto the sorbent. Moreover, according to the results of TEM analysis, the affinity purified recombinant gp053 formed noticeably shorter tubular structures (Supplementary Figure S2D,E).”

-   These two sentences are mutually exclusive. First it is stated that attempts to purify Hos-tagged variants gp053 were unsuccessful. In second sentence it is described that affinity purified gp053 formed shorter tubules.

Authors' response

We agree that these two sentences are mutually exclusive. The appropriate corrections have been made.

Reviewer 2 Report

The authors provide a study of the sheath protein gp053 from phage vB_EcoM_FV3. More specifically, they study its polysheaths, with the goal of enabling future applications.

Polysheaths of T4 have been well studied, unlike this one.

Although incremental, the authors present a solid characterisation of this protein and its ability to form polysheaths.

However, I have still several remarks.

1) in the mass spec results, the authors put exact number of residues that are missing from some of the proteins. However, they only describe that they have done peptide mass fingerprinting. Unless they have done mass determinations of the complete (degraded/truncated) protein bands, they cannot make such claims.

2) In image 6G and H, the fiber width appears to differ despite that these are similar constructs. After looking more carefully, this is because the magnification of 6G although very similar is different. Please put it on the same scale as all the other ones. 6I has the same issue but because it is so different it is not such a problem.

3) An important experiment is to check the specificity of the gold nanoparticle conjugation. The authors lack a negative control. The best control would be to make cysteine mutants and check if they abolish binding, ideally in all combinations. Alternatively, they might block cysteines on their polysheaths before incubation with gold nanoparticles.

Minor point:

line 63: "broad" should be "broaden"

Author Response

The authors provide a study of the sheath protein gp053 from phage vB_EcoM_FV3. More specifically, they study its polysheaths, with the goal of enabling future applications.

Polysheaths of T4 have been well studied, unlike this one.

Although incremental, the authors present a solid characterisation of this protein and its ability to form polysheaths.

However, I have still several remarks.

1) in the mass spec results, the authors put exact number of residues that are missing from some of the proteins. However, they only describe that they have done peptide mass fingerprinting. Unless they have done mass determinations of the complete (degraded/truncated) protein bands, they cannot make such claims.

Authors' response

We determined an exact number of residues that are missing from some of the proteins based on the results of qualitative and quantitative MS analysis of peptides.  On the other hand, we agree with reviewer's comment that mass determinations of the complete (degraded/truncated) protein bands would be a very suitable method to confirm the results of peptide mass fingerprinting. Nevertheless, as we have no potential to do mass determinations of the complete (degraded/truncated) protein bands currently, and in considering to comments from the reviewer, we have made appropriate corrections in the revised manuscript (see the Section 3.2 and the Discussion, lines 390‒392).

2) In image 6G and H, the fiber width appears to differ despite that these are similar constructs. After looking more carefully, this is because the magnification of 6G although very similar is different. Please put it on the same scale as all the other ones. 6I has the same issue but because it is so different it is not such a problem.

Authors' response

Figure 6 has been amended accordingly.

3) An important experiment is to check the specificity of the gold nanoparticle conjugation. The authors lack a negative control. The best control would be to make cysteine mutants and check if they abolish binding, ideally in all combinations. Alternatively, they might block cysteines on their polysheaths before incubation with gold nanoparticles.

Authors' response

We agree with reviewer's comment that it is very important to check the specificity of the gold nanoparticle conjugation. However, more detailed (and time-consuming) studies are needed to get accurate results. Thus, we are planning to construct cysteine mutants (together with a set of other gp053 mutants) in our next study.

Minor point:

line 63: "broad" should be "broaden"

Authors' response

“broad” has been amended accordingly.

Reviewer 3 Report

The article presented by Simoliunas et al. describes the purification of gp053 and its self-assembly forming nanotubes. Although this protein is similar to other tail sheath proteins, the purification protocol followed could ease the process to obtain protein compared with other similar proteins, and given the low sequence conservation, it could also expand the range of applications of the obtained nanotubes.

I would like to suggest some changes and point some questions that could be answered to improve the quality of the manuscript:

-       Although not the most common technique anymore, the authors could make use of their TEM images to generate a low resolution reconstruction of the nanotubes and fit their model of the protein into it. This technique was used in the past for many phage proteins successfully and would allow them to discuss the similarities and differences with other similar structures not only at a monomer level (i.e. T4, pyocins).

-       In the protein expression and purification sections, I miss some details for each different construct i.e. which one was expressed for 3h? were the rest left overnight?

-       On the other hand, the results section 3.2 says “20 to 37 degrees for three hours”. That’s confusing.

-       The authors could also discuss why they could not purify the protein using the His tag. The fact that it did not adsorb to the resin could be explained checking in the structure if the tag could be protected inside the protein. In any case, using ammonium sulfate was a very elegant alternative, eliminating unexpected interactions related with the tag.

-       Figure 2: “incubated at 30 degrees”. That is what I meant. Maybe a supplementary table with the exact conditions for each construct could help.

-       Figure 3D: to my eyes, the incubation with Urea also cleaned the background. Would that make sense? Could the authors comment the effect of each treatment for the background?

-       Figure 3C: the C in not visible.

-       The Supplementary figure S1 is difficult to read.

-       In my opinion, the S2 figure could be a main figure. The information is relevant.

-       Please check the manuscript for small typos.

Author Response

The article presented by Simoliunas et al. describes the purification of gp053 and its self-assembly forming nanotubes. Although this protein is similar to other tail sheath proteins, the purification protocol followed could ease the process to obtain protein compared with other similar proteins, and given the low sequence conservation, it could also expand the range of applications of the obtained nanotubes.

I would like to suggest some changes and point some questions that could be answered to improve the quality of the manuscript:

 -       Although not the most common technique anymore, the authors could make use of their TEM images to generate a low resolution reconstruction of the nanotubes and fit their model of the protein into it. This technique was used in the past for many phage proteins successfully and would allow them to discuss the similarities and differences with other similar structures not only at a monomer level (i.e. T4, pyocins).

 Authors' response

At this moment we do not have an appropriate experience to carry out the proposed analysis. Moreover, to our opinion, the I-TASSER generated model is not the best choice for fitting on 3D-reconstructions.

-       In the protein expression and purification sections, I miss some details for each different construct i.e. which one was expressed for 3h? were the rest left overnight?

Authors' response

We agree with reviewer that there are some obscurity describing the procedures of protein expression and purification. In response to comments, we have made the appropriate corrections (see section 2.2 Protein expression). In short, all constructs after induction were expressed at 30 °C for 3 hours. Only on purpose to get high yields of recombinant protein, 100‒200 ml of cells transformed with the appropriate plasmid were grown at 37 °C to an OD600 of 0.5, induced with 0.1 mM IPTG and incubated at 30 °C overnight.

-       On the other hand, the results section 3.2 says “20 to 37 degrees for three hours”. That’s confusing.

Authors' response

We agree that this sentence is confusing. Clarifying, it should be said that expression of recombinant gp053 in temperature range from 20 to 37 °C was performed during optimization procedures of expression of gp053. However, it was determined that three hours of incubation at 30 °C is an optimum for the expression of recombinant gp053. Thus, in order to reduce ambiguity, the sentence has been amended accordingly.

-       The authors could also discuss why they could not purify the protein using the His tag. The fact that it did not adsorb to the resin could be explained checking in the structure if the tag could be protected inside the protein. In any case, using ammonium sulfate was a very elegant alternative, eliminating unexpected interactions related with the tag.

Authors' response

This point was discussed in the Discussion section (see lines 392‒401). However, additional sentence has been inserted in revised manuscript (see lines 401‒403).

-       Figure 2: “incubated at 30 degrees”. That is what I meant. Maybe a supplementary table with the exact conditions for each construct could help.

Authors' response

As it was mentioned in the previous response, an expression of recombinant gp053 (as well as and in the cases of gp053 mutants) took three hours at at 30 °C. Thus, to our opinion, a supplementary table with the exact conditions for each construct is unnecessary.

-       Figure 3D: to my eyes, the incubation with Urea also cleaned the background. Would that make sense? Could the authors comment the effect of each treatment for the background?

Authors' response

We agree that it is possible that urea or trypsin could affect the rest of the bacterial proteins, which are left after purification and hereby clean the background. However, after more detailed analysis of a set of electron micrographs not presented in manuscript, we can not make exact conclusions.

-       Figure 3C: the C in not visible.

Authors' response

Figure 3 has been corrected accordingly.

-       The Supplementary figure S1 is difficult to read.

Authors' response

Figure S1 has been corrected accordingly

-       In my opinion, the S2 figure could be a main figure. The information is relevant.

Authors' response

Figure S2 has been transferred from supplementary material to the main manuscript (see Figure 2).

-       Please check the manuscript for small typos.

Authors' response

We made our best effort to correct the manuscript.